# Preparation of PI/PTFE–PAI Composite Nanofiber Aerogels with Hierarchical Structure and High-Filtration Efficiency

**DOI:** 10.3390/nano10091806

**Published:** 2020-09-10

**Authors:** Dawei Li, Huizhong Liu, Ying Shen, Huiping Wu, Feng Liu, Lanlan Wang, Qingsheng Liu, Bingyao Deng

**Affiliations:** 1Key Laboratory of Eco-Textiles (Ministry of Education), Nonwoven Technology Laboratory, Jiangnan University, Wuxi 214122, China; ldw@jiangnan.edu.cn (D.L.); hzliu16@163.com (H.L.); shenying1@sina.cn (Y.S.); whp17660453060@163.com (H.W.); fengliu_windy@163.com (F.L.); 15852776701@163.com (L.W.); qsliu@jiangnan.edu.cn (Q.L.); 2Kunshan Sunshinetex New Material Co., Ltd., No.417 Sanxiang Road, Industry zone, Kunshan 215300, China

**Keywords:** nanofiber-based aerogels, PTFE–PAI nanofiber, high-efficiency filter, PM2.5, hierarchical porous architecture

## Abstract

Electrospun nanofiber, showing large specific area and high porosity, has attracted much attention across various fields, especially in the field of air filtration. The small diameter contributes to the construction of filters with high-filtration efficiency for fine particulate matter (PM), however, along with an increase in air resistance. Herein, composited nanofiber aerogels (NAs), a truly three-dimensional (3D) derivative of the densely compacted electrospun mat, were constructed with the blocks of polytetrafluoroethylene–polyamideimide (PTFE–PAI) composite nanofiber and polyimide (PI) nanofiber. PI/PTFE–PAI NAs with hierarchically porous architecture and excellent mechanical properties have been obtained by thermally induced crosslink bonding. Results indicated that sintering at 400 °C for 30 min could complete the decomposition of polyethylene (PEO) and imidization of polyamic acid (PAA) into PI, as well as generate sufficient mechanical bonding between adjacent nanofibers in the NAs without extra additive. The well-prepared PI/PTFE–PAI NAs could withstand high temperature up to 500 °C. In addition, the filtration tests illustrated that the composite NAs had an excellent performance in PM filtration. More importantly, the filtration behavior could be adjusted to meet the requirements of various applications. The excellent thermal stability and high-filtration efficiency indicated its great potential in the field of high-temperature air filtration.

## 1. Introduction

Fibrous filters have the features of good filtration performance and a wide application range, and can effectively filter and remove fine particulate matter (PM) in the air, becoming the most important filtration technology to deal with air pollution. Filtration efficiency and pressure drop are the two key indicators to evaluate the performance of filter media. While the fibers capture PM, they also increase the drag force on the airflow. Therefore, achieving high efficiency and low resistance simultaneously still remains a big challenge.

In recent years, research has developed from various perspectives, e.g., considering single-fiber and fiber-accumulation structures, to solve the low physical interception efficiency existing in micro-fiber air filters. Nanofibers, with small diameters, high porosity, and high specific area, can be utilized to construct nanofibrous filters. Benefiting from the uniformity in fiber diameter, adjustable pore channels and stacked structures, nanofibrous filters have effectively captured PM in the air [1]. Moreover, the filtration efficiency has been improved by more than 70% compared with traditional filter media constructed by microfibers. However, severe problems still remain in the filter consisting of layered up nanofiber meshes, such as high air resistance, high energy consumption, poor mechanical properties, and easy peeling between layers [2,3,4,5]. Therefore, difficulties still exist in the preparation of ideal filter with electrospun nanofibers.

According to previous studies, the spatial distribution of fibers plays an essential role in the filtration efficiency and air resistance [6]. Thus, filtration performance of nanofibrous filters is closely related to the pore size and porosity of the fibrous material. Generally, three dimensional (3D) fibrous material with fluffy structure has a tortuous porous structure, which could capture PM in a longer path generated by the 3D dispersed fibers when the airflow passes through, increasing the filtration efficiency for PMs [7]. The pressure drop calculation formula is shown as follows:(1)$$\Delta p=\frac{\eta tU_{0}\left(64\alpha^{1.5}\left(1+56\alpha^{3}\right)\right)}{d_{f}^{2}}$$
where Δp is the pressure drop, *η* is the air viscosity, *U_0_* is the airflow velocity, *t* is the thickness of the filter, *d_f_* is the fiber diameter, and α is the solid phase volume fraction of the filter (0.006 < α < 0.3). For nanofiber materials with a fixed fiber diameter, the pressure drop is positively correlated with the solid phase volume fraction of the filter [8]. Compared with two-dimensional nanofiber membrane, 3D nanofiber material shows good inter connectivity, interface effect. The surface effect in the pore surface/interface micro-nano mass energy transfer process is significantly enhanced, forming a micro/nano scale transportation network and performing high-efficiency and selective transportation of the continuous media [9,10]. Therefore, it is quite important to design proper 3D structure of the nanofibrous filter to increase the porosity and thus reduce the solid phase volume of nanofiber assembly [11,12,13].

Up to now, methods utilized to construct 3D nanofibrous material via electrospinning include layer-by-layer assembling, auxiliary collecting, external field optimizing, as well as combining nanofiber and microfibers [14,15,16,17,18,19]. However, defects such as low strength, easy exfoliation, and in particular easy to collapse still occurred to the 3D nanofibrous materials. This is because the nanofibers deposit layer by layer on the collector, which makes it merely impossible for the nanofibers to penetrate into the assembly in the thickness direction [20,21,22]. When compressed or stretched in the thinness direction, the fluffy fiber agglomerates are quite easy to be compacted into a thin film or be torn into fragments due to the lack of effective support in thickness direction [23].

Fortunately, the 3D network reconstruction method provides a new strategy for constructing a 3D nanofiber assembly with improved porosity and mechanical properties [22,24,25]. During the process, electrospun nanofibrous mats are firstly immersed in the solution and cut into short fibers with a length of hundreds microns, which are then poured into the molds and frozen into solid. Subsequently, the solid containing nanofibers will be freezing dried, removing the solvent and leaving the nanofiber as a 3D assembly. Generally, an extra step of bonding or generate site-bonding, bringing about certain mechanical strength [21]. Thus, a reconstructed nanofiber aerogels (NAs) is prepared.

NAs possesses unique hierarchical structure, consisting of larger cell-like secondary pores with a diameter in the range of 30–100 μm and finer secondary pores, between entangled nanofibers, with the size of 2–5 μm [26,27]. Studies have indicated that the open pore structure can reduce the air drag force significantly. Polyimide (PI) NAs can filter out 99.9% of particulate matter, and its filtration resistance is only 177 Pa, while the filtration resistance of fiber membranes of the same material is as high as 1460 Pa [28]. In addition, the short nanofiber in the NAs are dispersed in an isotropic way, meaning that the fibers oriented uniformly in a 3D way [24]. The well-prepared NAs always possess super-elasticity and good resilience, which makes it easier to recover to its original dimension after compressing, twisting, and folding [22,29,30]. At the same time, the 3D network reconstruction method allows the hybrid composition of different fibers with various components, diameters, and forms [30,31,32]. Thus, combining the advantages of different nanofibers can facilitate further adjustment and optimization of the structure and function of NAs.

In this study, a novel strategy is demonstrated to create heating-resistant composite NAs by combining porous polytetrafluoroethylene–polyamideimide (PTFE–PAI) nanofiber with PI nanofiber, and it provide an easy way to generate hierarchical structure for high performance filtration. The advantage of this design is that the random-deposited nanofibers are reconstructed into elastic NAs with tunable densities and performance. Systematic tests were conducted to understand the structure and performance of the composite NAs, as well as its filtration efficiency. The mechanism of interaction between PTFE–PAI and PI nanofibers was elucidated to explain the enhanced stability of the composite NAs. For the purpose of investigating the effect of preparation conditions on the hierarchical structure of NAs, the morphology and microstructure of various NAs in different conditions was observed by SEM. Furthermore, the mechanical performance was fully discussed to evaluated the characteristic. Moreover, the pore size and filtration efficiency was investigated in detail.

## 2. Materials and Methods

### 2.1. Materials

PAI resin in H_2_O with a solid content of 30% was purchased from Nantong Bolian Material Technology Co. LTD (Nantong, China). Polyethylene (PEO) (Mw = 900,000) powder was supplied by Aldrich Chemistry (St. Louis, MO, USA). Polyamic acid (PAA) resin in N, N-dimethylacetamide (DMAc) with a solid content of 20% was purchased from Sumount Science and Technology Co., Ltd. (Hangzhou, China). Tert-alcohol (AR grade) was supplied by Aladdin Bio-Chem Technology Co., LTD (Shanghai, China). PTFE dispersion in H_2_O with a content of 60% was supplied by Zhonghao Chenguang Research Institute of Chemical Industry (Chengdu, China).

### 2.2. Preparation of PAA Nanofiber and PTFE–PAI@PEO Composite Nanofiber

The yellow viscous solution of PAA with a solid content of 20 wt% was loaded in a syringe equipped with a flat-tip s stainless spinneret with an inner diameter of 0.8 mm. A high voltage of 25 kV was applied on the spinneret. The electrospun fibers were deposited onto a grounded plate covered with aluminum foil. The collecting distance and feeding rate was 15 cm and 0.5 mL/h, respectively. The obtained nanofiber membrane was dried at 60 °C for 12 h in vacuum to remove the residual solvent.

On the other hand, PEO powder was firstly dissolved in the mixture of PTFE dispersion and distilled water along with vigorously stirring for 6 h, obtaining homogeneous PTFE–PEO solution with a ratio of PTFE:PEO = 10:1. Subsequently, varying amounts of PAI dispersion in H_2_O were added into the PTFE–PEO mixture and stirred for another 2 h to yield a series of homogeneous PTFE–PAI–PEO mixture. Then, the mixture was electrospun into nanofiber membrane via electrostatic spinning apparatus same as mentioned above. The obtained nanofiber was named as PTFE–PAI–PEO nanofiber. The feeding rate, applied voltage, and collecting distance was set as 0.8 mL/h, 18 kV, and 18 cm, respectively. The obtained PTFE–PAI–PEO nanofiber membrane was dried at 60 °C for 12 h in vacuum to remove the residual solvent. The formula of PTFE–PAI–PEO solution is listed in Table 1.

### 2.3. Fabrication of PI/PTFE–PAI Composite NAs

The schematic to fabricate PI/PTFE–PAI NAs is illustrated in Figure 1. PTFE–PAI–PEO nanofiber membrane and PAA nanofiber membrane was firstly cut into pieces (~1 × 1 cm), which were then immersed into the dispersing solvent (tert-butyl alcohol, TBA). Subsequently, the membrane pieces in a weight ratio of 3:7, 4:6, 5:5, 6:4 (PAA: PTFE–PAI–PEO) were further cut into short nanofibers using Hishear Dispersing Emulsifier (Fluko, Shanghai, China) at an operating speed of 15,000 rpm for 30 min. After homogenization, PTFE–PAI–PEO and PAA short-cut nanofiber would be mixed uniformly, forming well-blended nanofiber dispersion. Then, the dispersion was poured into the cylindrical molds, frozen at −20 °C for 6 h, and subsequently freezing dried under vacuum of 5 Pa for 48 h to sublimate the solvent thoroughly, obtaining the un-bonded NAs in the state of PAA/PTFE–PAI–PEO (named as un–NAs). Therewith, the thermal imidization process were completed via a staged sintering program: heating up to 100, 200 and 300 °C with a heating rate of 3 °C/min, keeping at each stage for 30 min. To generate crosslinking at the points of fibers, samples were further heated up to 350 °C, 400 °C and 450 °C with same manner. After the removal of PEO and imidization of PAA to PI, the mechanically bonded NAs were prepared in the state of PI/PTFE–PAI (named as PI/PTFE–PAI NAs).

### 2.4. Characterization

The surface and cross-section morphology of the nanofiber membranes and NAs was observed via a scanning electron microscope (SEM, Hitachi, Japan) at an acceleration voltage of 15 kV. The porosity (ε, %) was calculated via the following formula:(2)$$\varepsilon =\left(1-\frac{\rho_{bulk}}{\omega_{PTFE}\rho_{PTFE}+\omega_{PI}\rho_{PI}}\right)\times 100\%$$
where *ρ*_bulk_, *ρ*_PTFE_ and *ρ*_PI_ is the bulk density of NAs, the skeletal density of PTFE (2.2 g/cm^3^) and PI (1.1 g/cm^3^), respectively. *ω*_PTFE_ and *ω*_PI_ were the rate of PTFE and PI in the composite aerogels respectively. The shrinking rate after sintering was calculated the following equation:(3)$$\mathrm{Shrinkage}=\left(\frac{V_{0-}V_{1}}{V_{0}}\right)\times 100\%$$
where *V_0_* is the volume of nascent un–NAs and *V_1_* is the volume of the NAs after sintering.

Chemical structures of the composited NAs were examined by Fourier transform infrared spectrometer (FTIR, Thermo Fisher, Waltham, MA, USA) in the attenuated total reflection (ATR) ranging from 4000 to 400 cm^−1^. The thermal performance of samples (5 mg) were evaluated by thermo-gravimetric analyzer (TGA, TA Instruments, New Castle, DE, America) in the range of 40 °C–800 °C with a heating rate of 20 °C/min under N_2_ atmosphere.

The compressive property of PI/PTFE–PAI NAs were tested by compressing to a strain of 80% at the rate of 5 mm/min via the computerized tensile testing machine (HY-940FS, Hengyu, Shanghai, China). The filtration performance was evaluated by measuring the filtration efficiency and pressure drop of the NAs samples via Filtration Material Comprehensive Performance Test Bench (LZC-H, Suzhou Huada, China) at the flow rate of 84 L/min.

## 3. Results and Discussion

### 3.1. Preparation PAA and PTFE–PAI@PEO Nanofiber

#### 3.1.1. PAA and PI Nanofiber

PAA was chosen as the electrospun matrix due to its good spinnability and thermal performance. Figure 2 shows the SEM micrographs of PAA nanofibers under different electrospinning parameters. Separately, Figure 2a–c shows the effect of the feeding rate on the fiber morphology and diameter. When the feeding rate was 0.5 mL/h, the PAA nanofiber revealed smooth surface with uniform diameter of (3.4 ± 0.5) × 10^2^ nm. Increasing the spinning speed to 0.8 mL/h (Figure 2b) and 1.2 mL/h (Figure 2c) would result in uneven diameter and thick nubs. The PI nanofiber obtained at the rate of 0.5 mL/h shows the best morphology, which was fixed as the default for further experiments. Moreover, increasing the voltage (from 10 kV to 15 kV) applied on the spinneret would favor the sufficient stretching of polymer jets, leading to the decrease of diameter and removal of ribbon-like structure (Figure 2d). Uniform PAA nanofiber with concentrated diameter was obtained at the voltage of 15 kV (Figure 2a). Further increasing of voltage to 20 kV would cause unstable whipping of the spinning jets. The spinning solution would be torn into more branches, which would seduce the entanglement bonding and an uneven distribution of fiber diameter (Figure 2e), (5.2 ± 1.8) × 10^2^ nm).The collecting distance performed an opposite function on the nanofiber formation to voltage (Figure 2f,g). In summary, when the feed rate, applied voltage, and spinning distance was 0.5 mL/min, 15 kV, and 18 cm, respectively, the obtained PAA nanofibers showed a uniform structure and similar size (Figure 2a). The PAA nanofiber membrane was further sintered to complete the imidization through a step-processing, obtaining PI nanofiber (Figure 2h,i), which possessed regular morphology and uniform diameter. No significant morphological change could be observed (Figure 2i).

#### 3.1.2. PTFE–PAI–PEO and PTFE–PAI Nanofiber

PTFE possesses decent chemical resistance and heat-resisting property, which was usually utilized to prepare PTFE coated high temperature filter bag. Due to the discontinuity of PTFE NPs in the dispersion, it is impossible to prepare PTFE nanofiber directly via electrospinning. In previous studies, a sacrificial polymer is always picked to help generate continuous nanofiber with PTFE NPs embedded in the matrix of polymer, which will then be burned to remove the sacrificial polymer and melt the PTFE NPs. The molten PTFE NPs will become soft and fuse to stick together, forming a chain of PTFE NPs or, in other words, a PTFE nanofiber. However, in the sintering process, severe shrinkage occurred to the PTFE nanofiber as molten PTFE NPs would fuse with each other after removing the matrix of PEO [33,34]. On the other hand, the water-soluble PAI resin belongs to a series of high molecular polymers with regular and alternating arrangement of imide rings and amide bonds. Thanks to the heat-resistant aromatic heteroimine groups and flexible amide groups, fiber made of PAI also shows excellent heat resistance, dielectric properties, mechanical properties and chemical stability [35,36]. In addition, the water solubility of PAI resin make it possible to prepare hybrid nanofiber accompanied with PTFE–PEO through electrospinning.

Herein, PAI was incorporated into the PTFE–PEO solution to prepare PTFE–PAI–PEO nanofiber via electrospinning. Figure 3 shows the SEM images of PTFE–PAI–PEO nanofiber generated from the solution of different formula (the flow rate of the spinning solution, applied voltage, and spinning distance was set as 0.8 mL/h, 18 kV, and 18 cm, respectively). We found that uniform nanofiber could be collected only when the weight ratio of PTFE–PAI was 3:7 (Figure 3b–e,b’–e’). Furthermore, we made a change of the volume of H_2_O to adjust the concentration of the spinning solution. The ratio of PTFE: PEO:H_2_O was set as 10:1:4, 10:1:4, 10:1:4, 10:1:4, and 10:1:4 (Table 1), respectively. When the mass ratio of H_2_O was as low as 4, uneven fibers with wide diameter distribution could be observed, while NPs could be seen distributed uniformly on the surface (Figure 3h). Fiber glued together at the cross-points were collected while increasing the ratio of water to 12 (Figure 3c). Uniform PTFE–PAI–PEO nanofibers could be obtained when the ratio ranging from 19 to 29 (Figure 3c–e). The diameter decreased from (7.2 ± 3.2)·× 10^2^ nm to (3.2 ± 1.2)·× 10^2^ nm. Further increasing the ratio of water to 39 would generate dilute solution with poor spinnability, resulting in beads and droplets in the membrane (Figure 3g). After sintering at 400 °C for 30 min, the PTFE–PAI–PEO nanofiber was converted to PTFE–PAI nanofiber, accompanied by the changing of fiber diameter and surface morphology. Sintering process would increase the diameter for those fiber with smaller diameter (Figure 3c,c’,e,e’), but oppositely decrease the diameter for thick fibers (Figure 3b,b’,d,d’). Additionally, nanoscale pores were observed on the surface of PTFE–PAI nanofibers, especially in Figure 3c’. That could be attributed to the decomposition of PEO, which left voids in the body of PAI–PTFE nanofiber [37]. No nanoscale pores could be left after the removal of PEO for the PTFE–PEO nanofiber, due to the molten PTFE NPs were tightly held together by the cohesion force. However, in the PTFE–PAI–PEO nanofiber, the existence of PAI would hold the PTFE NPs and maintain the voids after the removal of PEO. Thus, the porous structure in PTFE–PAI nanofiber was generated, giving it larger specific surface area and higher porosity. Herein, PTFE–PAI–PEO nanofiber derived from the solution formula of PTFE: PEO: H_2_O = 10:1:9 and PAI: PTFE = 3:7 was chosen for further study.

### 3.2. Fabrication of PI/PTFE–PAI NAs

The preparation procedure of composite NAs is schematically illustrated in Figure 1. PAA (0.5 mL/min, 15 kV, and 18 cm) and PTFE–PAI–PEO (hybrid–3) nanofibers were prepared as mentioned above. The two kinds of nanofibers with different ratios (2:8, 3:7, 4:6, 5:5, 6:4) were firstly broken into short nanofibers in TBA solution by the high-shear homogenizer at the rate of 15,000 rpm for 30 min. Thereafter, the suspension containing shortened PAA and PTFE–PAI–PEO nanofibers was transferred into a cylindrical mold and then freeze dried to form a un-bonded NAs (un–NAs). The formation mechanism of the 3D structure could be found in our previous publication [20]. Finally, the un-bonded NAs was thermally sintered at high temperature (300 °C, 350 °C, 400 °C and 450 °C.) to form a robust 3D structure. During the sintering process, PEO was heated to decomposition, leaving porous PTFE–PAI nanofiber. At the same time, PAA was transformed into PI by thermal imidization. Thus, the PI and PTFE–PAI nanofiber formed the open-pore network of composite NAs. The PTFE NPs in the PTFE–PAI nanofiber would serve as the binding agent for the shortened nanofibers in the 3D NAs. PI was a necessary component to improve the mechanical integrity and reduce shrinking rate of the composite NAs. Various conditions were investigated to make clear the formation mechanism of composite NAs, including the PAA/PTFE–PAI–PEO ratio, nanofibers loading contents, as well as the sintering temperature.

#### 3.2.1. Influence of Weight Ratio of PAA: PTFE–PAI–PEO Nanofibers

The un-sintered PAA and PTFE–PAI–PEO nanofiber membranes were utilized to prepare PAA/PTFE–PAI–PEO NAs. The nanofiber mats were cut into pieces and then immersed into TBA, which were further homogenized into single short nanofibers by a high shear homogenizer to give a well-blended nanofiber dispersion. The disorganized nanofibers throughout the dispersion formed a 3D isotropic network, which could be maintained then freeze-drying to sublimate the solvent, forming un-bonded PAA/PTFE–PAI–PEO NAs (named as un-NAs). Furthermore, extra bonding was required to strengthen the un-NAs. Here, we took advantage of the melting feature of PTFE NPs to bond the adjacent fiber at the cross-points. Excellent elasticity and mechanical strength was achieved after thermal bonding. The influence of fiber ratio on the morphology and structure of PI/PTFE–PAI NAs was investigated. Figure 4a displayed the photograph of PI/PTFE–PAI NAs with a fiber content of 1% of different fiber ratios, sintering at 400 °C for 30 min. As can be seen from the photos (Figure 4a,b), visual color changed from white to light yellow with increasing PAA nanofiber amounts. After sintering, the color of un-NAs turned deep yellow. The NAs of 2:8 suffered from severe shrinkage, and even lost its cylindrical shape. SEM images indicates that the highly fluffy texture became densely compacted structure after sintering (Figure 4c_1_–c_4_). With the increase ratio of PAA nanofibers, the shrinkage of the NAs after sintering tend to be minimized. It could be confirmed from the SEM images of PI/PTFE–PAI NAs (Figure 4c_4_–g_4_). The PTFE–PAI nanofibers served as the binder, sticking the PI nanofiber and PTFE–PAI nanofiber themselves. The cohesion of PTFE–PAI nanofiber tended to pull themselves together, inducing shrinkage of the NAs. As shown in Figure 4h, for the NAs of same content and sintering temperature, when the ratio of PAA: PTFE–PAI–PEO increased from 3:7 to 4:6, the shrinkage of NAs dropped significantly. However, as the content of PAA continued to increase, the decreasing trend of volumetric shrinkage began to be stable, especially when PAA: PTFE–PAI–PEO was between 5:5 and 6:4. The increasing in PAA amount could improve the stability due to the existence of PI nanofibers blocked the contact sites between PTFE–PAI nanofibers, resulting in the ease of the agglomeration effect. When the ratio of PI increased to 5:5 and 6:4, the shrinking rate could be controlled under 25% (Figure 4h). Porosity of the NAs was also calculated, which indicated the NAs with a weight ratio of 5:5 possessed the highest porosity, up to 99%, after sintering at 400 °C (Figure 4i).

#### 3.2.2. Influence of the Fiber Contents

The effect of nanofiber solid content on the morphology and structure of PTFE–PAI/PI nanofiber aerogel was investigated. Figure 5 displays the morphological changes of NAs with different fiber contents at the weight ratio of PAA: PTFE–PAI–PEO = 5:5, sintering at 400 °C for 30 min. After sintering, the fragile NAs turned to be robust. It can be seen from the Figure 5a,b that, as the nanofiber content increased, the deformation of the NAs gradually decreased. When the nanofiber content was 0.5%, the mesh of the PI/PTFE–PAI NAs was relatively sparse (Figure 5c_1_–c_3_) and the pore diameter on the cellular wall was quite large. As the contents increased, the loosely linked structure turned denser gradually, along with the increasing of the small pores on the cellular wall, making the overall skeleton structure of NAs stronger. When the fiber content increased to 2%, the mesh in the PI/PTFE–PAI nanofiber aerogel was extremely dense (Figure 5f_1_–f_3_). As shown in Figure 5g, NAs of low contents suffered from more deformation caused by sintering, reaching 28% for 0.5%. Increasing the NAs contents from 0.5% to 2.0%, the loss in sample size slowly decreased and gradually tended to be stable after reaching 1.5% and 2.0%. Along with the increase of nanofiber contents, a slight decline in the porosity was detected (Figure 5h). More nanofiber would occupy more space in the NAs, which necessarily caused the reduction of porosity.

#### 3.2.3. Influence of the Fiber Contents

As the sintering process plays a critical role in the formation of mechanical strengthened NAs, it is quite necessary to make clear the impact of sintering temperature. Figure 6 shows the photos and SEM images of PI/PTFE–PAI NAs treated at different temperatures with a fiber content of 1.0%, PAA/PTFE–PAI@PEO ratio of 5:5. Before sintering, the PTFE–PAI–PEO nanofiber illustrated a straight and short rod structure with a bigger diameter and spherical protrusions on the surface. On the contrary, the curly PAA nanofiber with a smaller diameter and smooth surface dispersed among the PTFE–PAI–PEO fibers (Figure 4f_1_,f_2_). These two types of nanofibers were stacked randomly and intertwined to form an aerogels structure. However, there was no bonding between the fibers at this time, as a consequence extra crosslinking was necessarily needed. As shown in Figure 6a, as the sintering temperature increased, the color of NAs gradually changes from light yellow to deep yellow, and the size declined slightly. When treated under high temperature, the PTFE–PAI–PEO nanofiber turned to PTFE–PAI nanofiber after the decomposition of PEO, leaving numerous nano-voids on the surface. The PTFE–PAI nanofiber would also become partly molten and soften at high temperature, leading to the bending of the porous fibers and the adhesion of PTFE–PAI nanofiber to adjacent nanofibers (Figure 6b’–d’). An increase in the treating temperature could generate larger deformation and bending angles, along with the spreading of contact sites (Figure 6e,e’), increasing the stability of the NAs structure. However, as the sintering temperature reached 450 °C, due to the high temperature, the adhesion between the fibers became too serious, and the pore structure built by the nanofiber skeleton inside the NAs was blocked and destroyed.

### 3.3. Chemical Analysis

The sintering process not only caused the decomposition of PEO, but also completed the imidization of PAA. Physicochemical analysis was conducted to study the physical and chemical change of NAs before and after thermal treatment. Figure 7a displays the FTIR spectra of PI/PTFE–PAI NAs after sintering at different temperatures. Before sintering, weak absorption peaks at 1777 cm^−1^, 1380 cm^−1^, and 725 cm^−1^ were observed, which corresponded to the Ⅱ, Ⅲ and Ⅳ bands of the imide, respectively. After sintering at different temperatures, the absorption peaks at these locations became more obvious. That could be explained as, in the PAI molecule before sintering, the amount of imide bonds in the entire NAs was relatively low, representing weak absorbance peaks. After sintering, due to the cyclization reaction of carboxyl and amino groups in PAAs and PAI. Thus, the characteristic peaks originally representing C=O stretching vibration at 1652 cm^−1^ (i.e., amide I band) and 1540 cm^−1^ symbolizing C–N stretching vibration (i.e., amide II band) disappeared after heat treatment, generating more imide groups. As only a small amount of PEO was added into the PTFE–PAI–PEO nanofiber, the characteristic peaks corresponding to PEO was quite weak in the curve before sintering, which then disappeared after sintering, indicating that the decomposition of PEO was completed. The characteristic peaks of PTFE appeared around 1145, 1200 and 628 cm^−1^ before and after sintering, indicating that no chemical change occurred to the PTFE component.

### 3.4. Thermal Performance of PTFE–PAI/PI NAs

The reasonable setting of the sintering temperature has a great influence on the NAs strengthening process. It should be high enough to complete the decomposition of PEO, the imidization of PAA into PI, as well as the molten of PTFE NPs. Additionally, a too high temperature would generate severe shrinkage and damage on the structure of NAs. Only appropriate sintering temperature could maintain the pore structure of the NAs and preserve all three components of PTFE, PAI, and PI. Figure 7b shows the DSC curves of PTFE–PAI–PEO nanofibers and PTFE–PAI nanofiber after heating treatment at 400 °C. Initially, the fibers showed endothermic peaks at 58, 167 and 343 °C, which corresponded to the melting temperatures of PEO, PAI and PTFE. However, the general melting point PTFE is 327 °C, which is lower than that of the PTFE–PAI–PEO. That maybe due to the embedding of PEO matrix would level up the melting point of PTFE, which was simultaneously undergoing gradual decomposition. The DSC curve of the sintered PTFE–PAI nanofibers only showed a melting peak corresponding to the PTFE at 327 °C, which showed a slightly decrease without the support of PEO component. That made it clear that the sintering process at 400 °C successfully completed the decomposition of PEO. Moreover, PAI would not undergo softening and melting behavior within 400 °C.

Thermogravimetric analysis (TGA) was conducted to study the thermal stability of the prepared NAs. Figure 7c reveals the thermal weight loss curve of PAI polymer and PI/PTFE–PAI NAs. For the pure PAI polymer without sintering, two weight loss stages were observed. The first one started from 105 °C was caused by the loss of residual solvent and the cyclization dehydration of PAI. The second one started at 445 °C indicated the decomposition of PAI polymer. PAI still had a residual weight of more than 30% when the temperature raised to 800 °C. The weight loss rate of the sintered PI/PTFE–PAI NAs at 505 °C is only 5%, which is mainly caused by the evaporation residual H_2_O, a small amount of residual PEO and partial cyclization dehydration in the molecular chain of PI and PAI. After that, as the temperature increased, PTFE, PAI, and PI began to gradually decompose, and finally the residual weight at 800 °C was about 43%. It was proved that the PI/PTFE–PAI NAs exhibited good thermal stability under 500 °C.

### 3.5. Mechanical Properties

The mechanical properties of all composite NAs were tested under compression to a strain of 80%. No sample fractured during the measurement. Figure 8a,c,e shows the compressive stress-strain curve of PI/PTFE–PAI NAs of different weight ratio, solid contents, and sintering temperature, respectively. As shown in Figure 8a,b, changing the weight ratio of PAA: PTFE–PAI–PEO nanofiber would induce slight change in the compressive strength at 80% strain. When the ratio was 6:4, the composite NAs achieved the maximum strength to 8.0 kPa. However, no significant difference was observed among different ratios. Apparently, the solid contents had a significant effect on the compressive behavior. An increase in the contents from 0.5% to 2.0% leveled up the strength from 1.2 kPa to 13.7 kPa (Figure 8c,d). That was because when the nanofiber solid content was too low, the 3D structure of the NAs was loosely linked by a small amount of nanofiber. The cellular wall with large pores on it showed small thickness, which had a poor compressive property. An increase in the solid contents from 0.5% to 2.0% would therefore bring a tight arrangement of nanofibers and a thicker cellular wall. The skeleton structure of the entire NAs grew stronger, resulting in an enhanced mechanical performance.

As the sintering process is the key to generate mechanical bonding between adjacent fibers, the treating temperature plays a critical role in the mechanical performance. Figure 8e,f illustrates the mechanical performance of NAs sintered under different temperature. After sintering at 300 °C for 30 min, the compressive stress at 80% strain of NAs was 2.1 kPa. As the sintering temperature increased, the compression performance gradually improved from 2.2 to 7.3 kPa at 450 °C. PEO decomposition was started from 180 °C, and the imidization process could be completed at 300 °C. However, the PTFE started to be molten at 327 °C. NAs treated at 300 °C could not melt the PTFE NPs and form bonding between nanofibers. Increasing in the temperature above 350 °C could induce the soften and adhesion of PTFE NPs at the contact sites, which could be confirmed in Figure 6. The PTFE NPs melted to compensate for the weak joints between the PTFE particles and PAI components after the decomposition of PEO, and bonded with the surrounding fibers to form a stable network, thus the mechanical property was improved.

### 3.6. Filtration Performance

NAs constructed by the short cut nanofiber possess a high porosity and open-pore network, which can effectively capture PMs in the air. Herein, the porous structure and filtration efficiency of composite NAs was studied. Figure 9a–d shows the pore size distribution of PI/PTFE–PAI NAs with different solid contents of 0.5%, 1.0%, 1.5% and 2.0%, respectively. When the nanofiber content was 0.5%, the average pore size was ~3.7 μm. When the content increased to 1%, the pore size dropped to ~3.4 μm. An increase in solid content to 2% would further reduce the pore size to ~1.8 μm. As the fiber content increased, the pores between the nanofibers gradually decreased and the pore size distribution showed a more concentrated mode. It is well known the pore size displayed great influence in the filtration performance due to the physical blocking.

The high porosity and specific surface area of NAs is conducive to improve the filtration performance. Figure 9e,f and Table 2 shows the filtration performance and specific value of the composite NAs of different fiber solid contents. It could be seen from Figure 9e that the filtration efficiency of NAs with different fiber contents all showed an upward trend as the size of PM increased. A linear dependence between the solid contents and pressure drop was observed upon measurement the pressure changed as a function of the porosity. The solid contents rose from 0.5% to 2.0% resulted in the decrease of porosity from 98.52% to 96.17%, which subsequently increased the pressure drop from 180 Pa to 440 Pa. The filtration mechanism of fine particles by a single was explained previously [26]. The hierarchical structure played a critical role in the high-filtration efficiency. As it is known, three major mechanisms for particle capture are via diffusion, interception and impact. Increasing the specific surface area by using porous nanofibers has a great impact on the diffusion-driven capture. Alternatively, smaller pores formed by nanofibers can enhance the efficiency of particle capturing driven by interception and impact mechanism. The PI/PTFE–PAI NAs possessed the nanofibrous skeleton and high-porous structure, as well as porous surface of PTFE–PAI nanofiber. Which could facilitate the pass through of air flow while blocking more particles. For PM_2.0_, all NAs showed excellent filtration performance with an efficiency above 99.34%, up to 99.98% for 2.0% contents. For smaller particle size, such as PM_1.0_ and PM_0.5_, NAs with lower density had witnessed a drop in capturing efficiency, the NAs of 0.5% and 1.0% contents could only capture 91.27%, 95.51% of PM_0.5_, respectively. However, the NAs with 2.0% nanofibers showed a high efficiency of 99.19% and 97.35% for PM_0.5_ and PM_0.3_, respectively.

## 4. Conclusions

In summary, we put forward a novel strategy to prepare a high performance filter for high temperature flue gas filtration. This material possesses good thermal stability, and also shows excellent filtration properties. To achieve the excellent performance of the composite NAs filter, three high performance polymers (PI, PTFE, PAI) were selected as the building blocks, which were constructed into high-fluffy PI/PTFE–PAI NAs by combining electrospinning and a deconstruction-reconstruction technique. Due to the melting behavior of PTFE NPs during sintering, fixed points were formed between PTFE–PAI and PI nanofiber, giving the composite NAs certain strength. Additionally, the imidization of PI and PAI also improved the thermal stability up to 500 ℃, making it possible to stand the high-temperature of flue gas. Benefiting from the high specific area of PTFE–PAI nanofiber and the hierarchical structure of NAs, the composite NAs filters showed balanced filtration performance, which could be adjusted to meet the requirements of various application fields. Nowadays, the volume production of electrospun nanofiber has been realized with the help of needleless electrospinning, which can facilitate the industrialization of the composite NAs through this novel and simple strategy. It is believed to be an important candidate in various applications including respirator protection, indoor air purification, as well as PM_2.5_ pollutions control.

## Figures and Tables

**Figure 1 nanomaterials-10-01806-f001:**
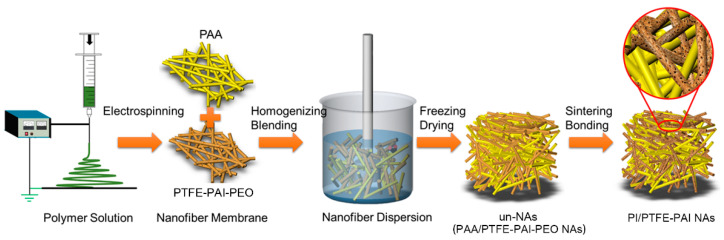
Schematic of preparation of PI/PTFE–PAI NAs via reconstruction strategy. PI, polyimide; PTFE, polytetrafluoroethylene; PAI, polyamideimide.

**Figure 2 nanomaterials-10-01806-f002:**
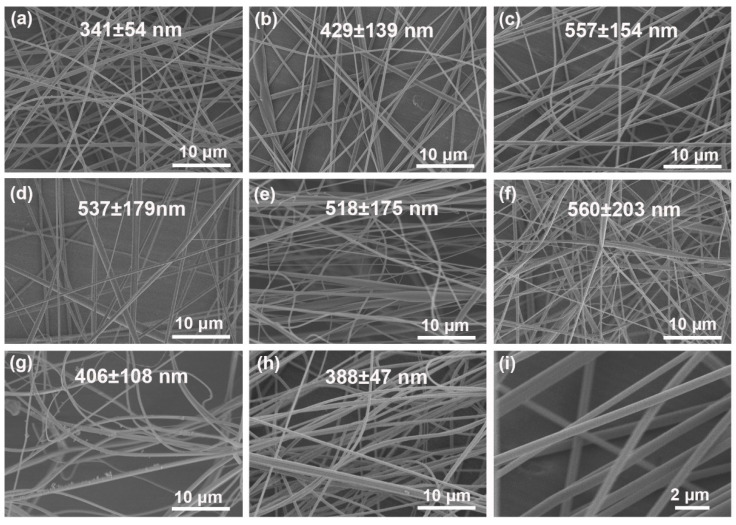
SEM and diameter of PAA nanofiber under different electrospinning parameters. The feeding rate, voltage, and collecting distance was (**a**) 0.5 mL/h, 15 kV, 18 cm; (**b**) 0.8 mL/h, 15 kV, 18 cm; (**c**) 1.2 mL/h, 15 kV, 18 cm; (**d**) 0.5 mL/h, 10kV, 18 cm; (**e**) 0.5 mL/h, 20 kV, 18cm; (**f**) 0.5 mL/h, 15 kV, 10cm; (**g**) 0.5 mL/h, 15 kV, 26cm; (**h**,**i**) 0.5 mL/h, 15 kV, 18 cm, after sintering, respectively.

**Figure 3 nanomaterials-10-01806-f003:**
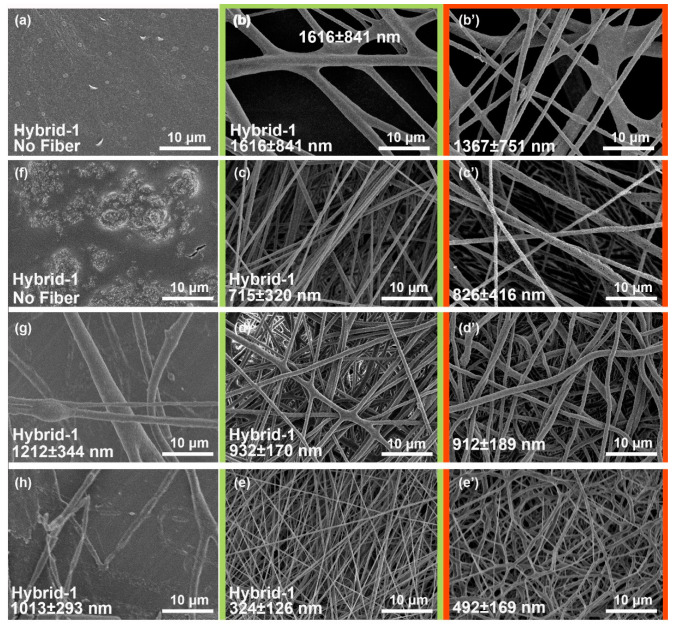
SEM of PTFE–PAI–PEO nanofiber with different parameters of spinning solution. (**a**) hybrid–1, (**b**) hybrid–2, (**c**) hybrid–3, (**d**) hybrid–4, (**e**) hybrid–5, (**b’**–**e’**) PTFE–PAI nanofiber after sintering transferred from b, c, d, e, respectively; (**f**) hybrid–6, (**g**) hybrid–7, (**h**) hybrid–8.

**Figure 4 nanomaterials-10-01806-f004:**
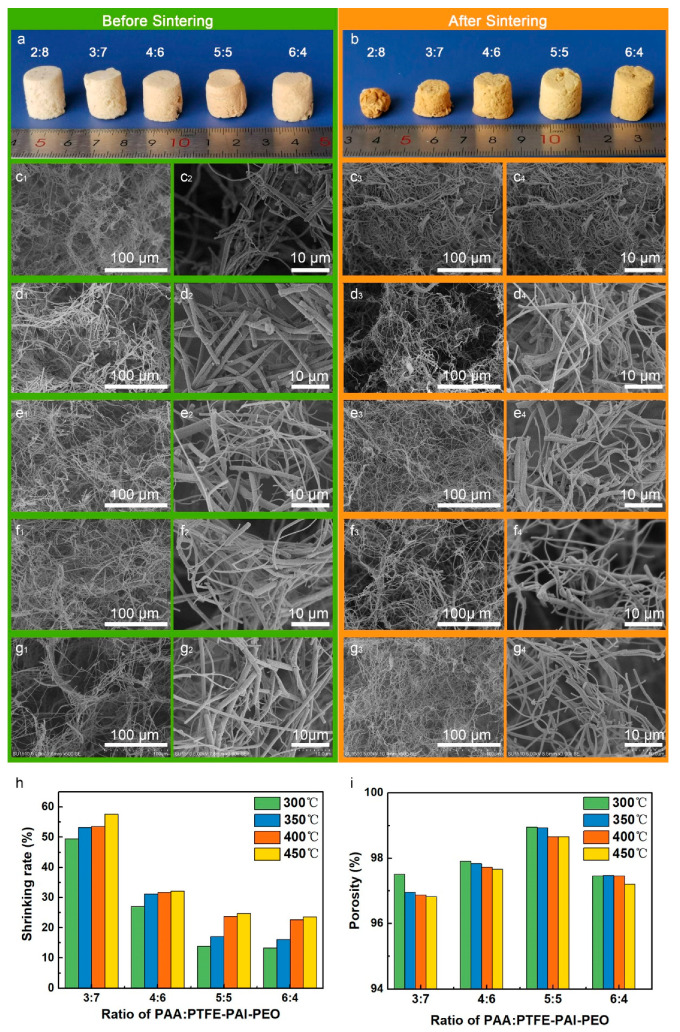
Photographs of the (**a**) un-bonded nanofiber aerogels (NAs) (**b**) and PI/PTFE–PAI NAs after sintering with different weight ratio of PAA:PTFE–PAI–PEO nanofibers. SEM images of NAs before and after sintering of different ratios of PAA: PTFE–PAI@PEO nanofiber: (**c_1_**–**c_4_**), 2:8; (**d_1_**–**d_4_**), 3:7; (**e_1_**–**e_4_**), 4:6; (**f_1_**–**f_4_**), 5:5; (**g_1_**–**g_4_**), 6:4. The fiber content was 1.0%, and sintering was conducted at 400 °C for 30 min; (**h**) shrinkage and (**i**) porosity of different PI/PTFE–PAI NAs after sintering at different temperature.

**Figure 5 nanomaterials-10-01806-f005:**
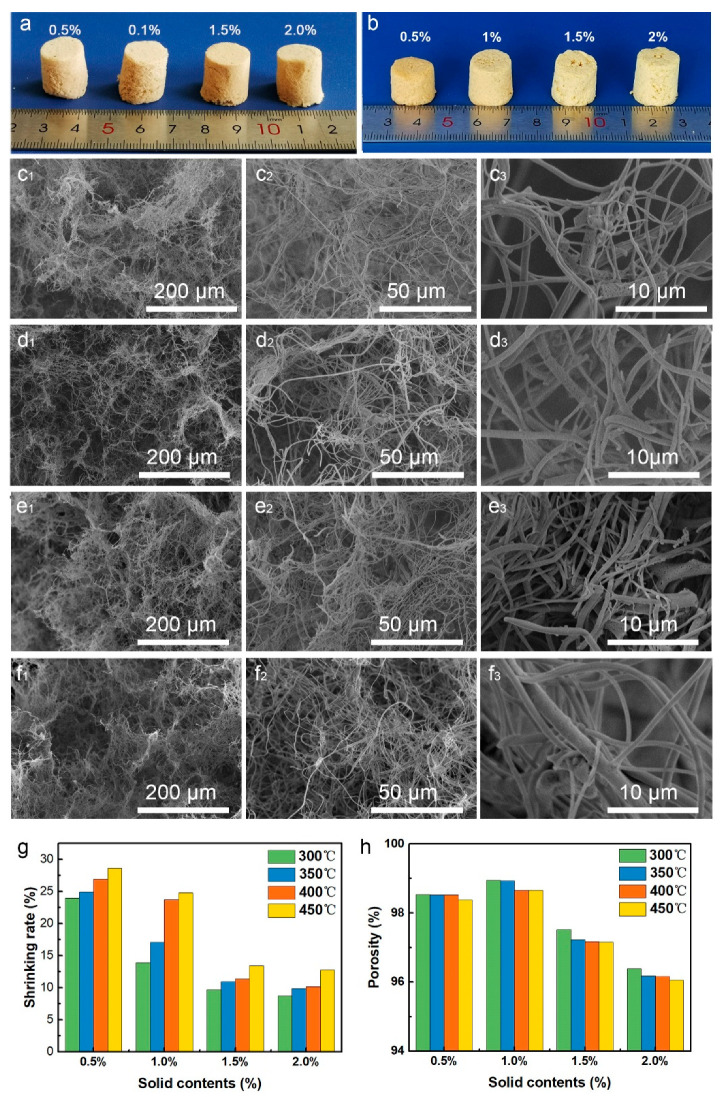
Photograph of un-NAs (**a**) and PI/PTFE-PAI NAs (**b**) and SEM images of PI/PTFE–PAI NAs with different fiber content; (**c_1_**–**c_3_**) 0.5%; (**d_1_**–**d_3_**) 1%; (**e_1_**–**e_3_**) 1.5%; (**f_1_**–**f_3_**) 2% as the ratios of PAA: PTFE–PAI–PEO nanofiber was 5:5 and the sintering temperature was 400 °C; (**g**) shrinking rate and (**h**) porosity of PI/PTFE–PAI NAs of different contents after sintering at different temperature.

**Figure 6 nanomaterials-10-01806-f006:**
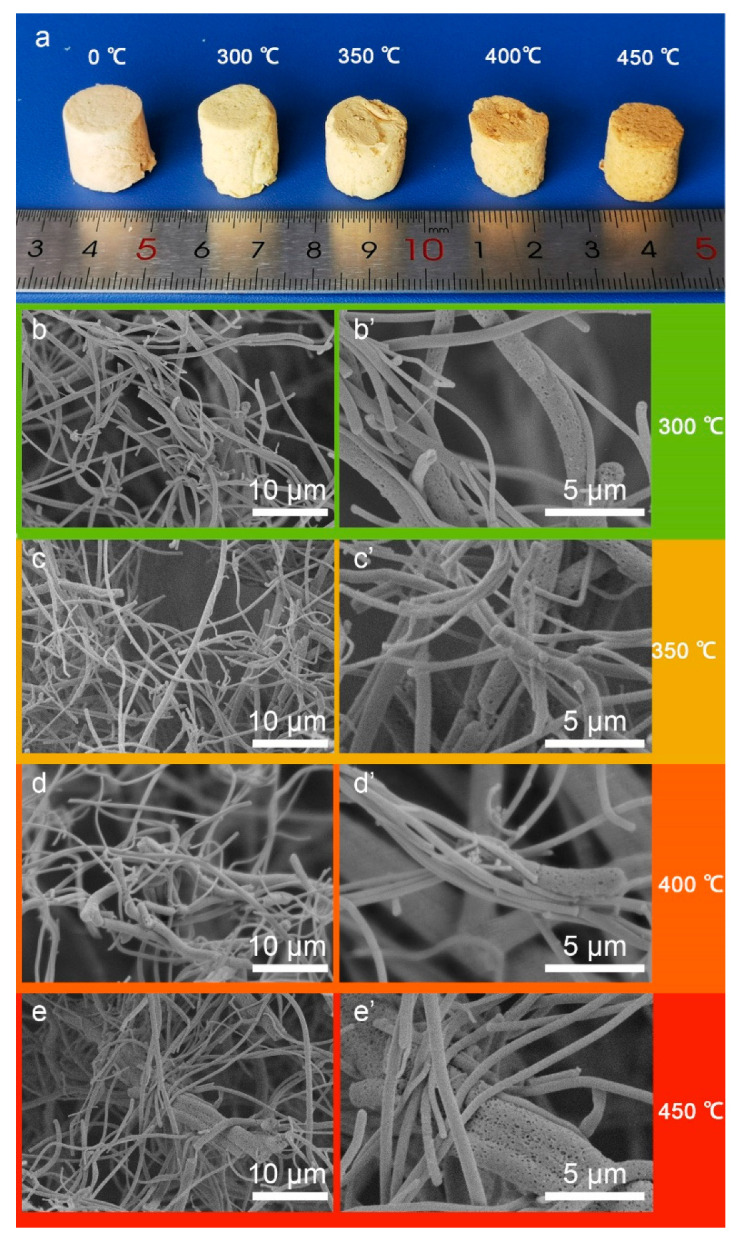
(**a**) Photograph and SEM images of PI/PTFE–PAI NAs at different sintering temperature as the ratios of PAA/PTFE–PAI@PEO was 5:5 and the content of fibers was 1%: (**b**,**b’**) 300 °C; (**c**,**c’**) 350 °C; (**d**,**d’**) 400 °C; (**e**,**e’**) 450 °C.

**Figure 7 nanomaterials-10-01806-f007:**
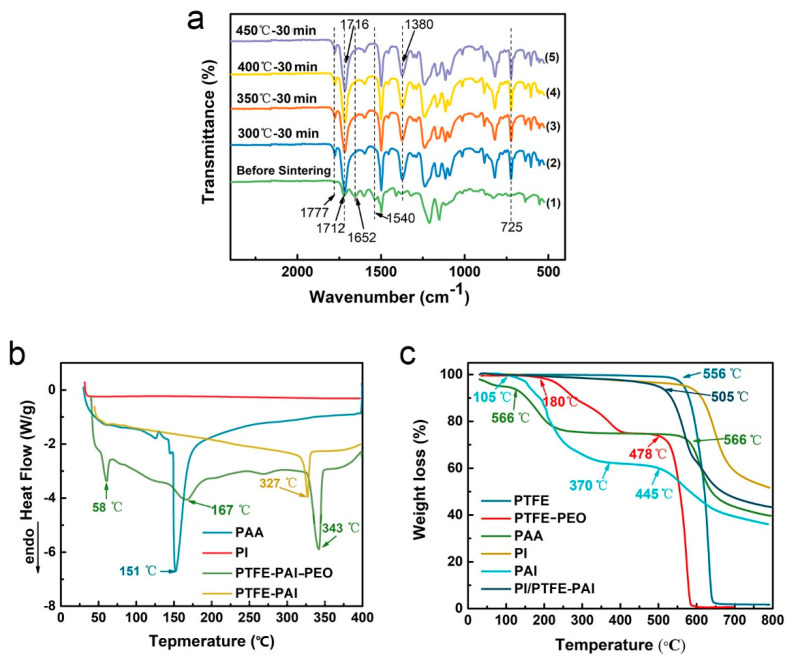
(**a**) FTIR of PI/PTFE–PAI NAs sintered at different temperatures as PAA/PTFE–PAI was 5:5 and the fiber content was 1.0%; (**b**) the DSC curve of PAA, PI, PTFE–PAI–PEO, and PTFE–PAI nanofiber; (**c**) the TG curves of PTFE NPs, PAI polymer, and PTFE–PEO, PAA, PI, PI/PTFE–PAI nanofiber.

**Figure 8 nanomaterials-10-01806-f008:**
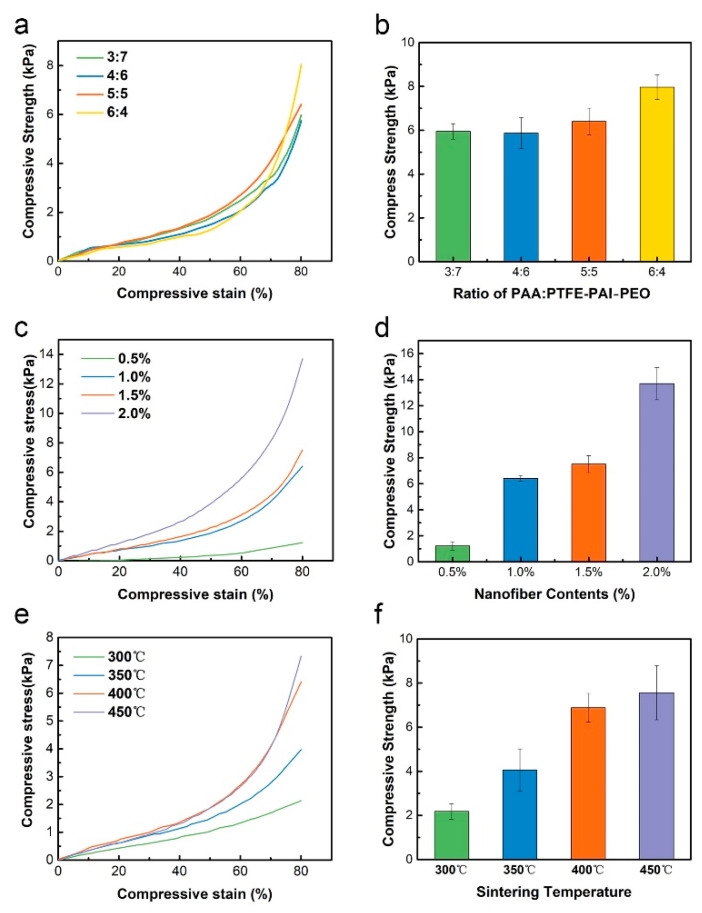
The compressive stress–strain curves and compressive strength at 80% of PI/PTFE–PAI NAs. (**a**,**b**) NAs of different weight ratio; (**c**,**d**) NAs with different solid contents; (**e**,**f**) NAs sintered at different temperatures.

**Figure 9 nanomaterials-10-01806-f009:**
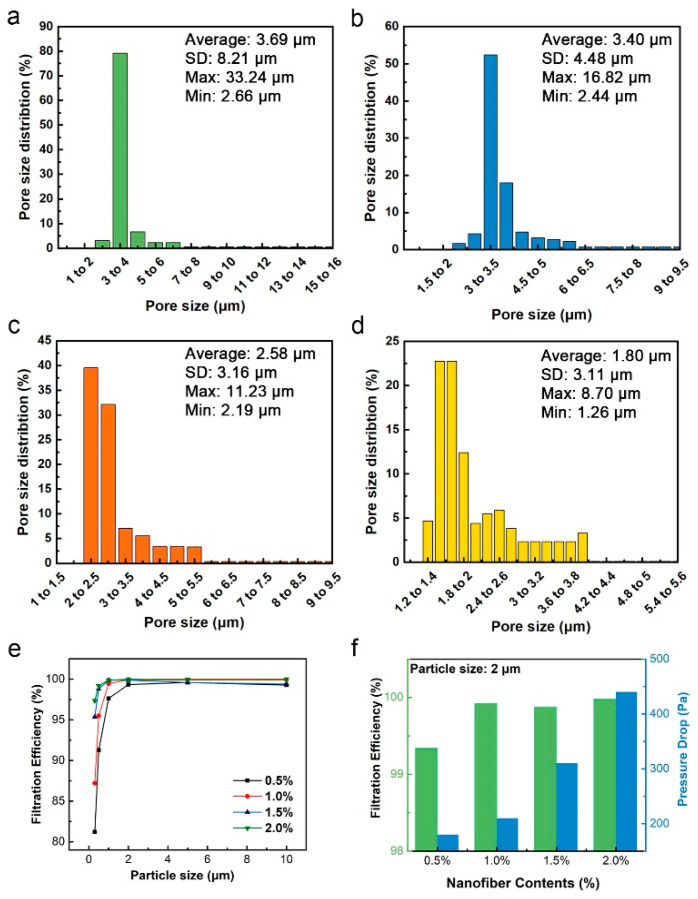
The pore sizes and pore sizes distribution of PI/PTFE–PAI NAs with different fiber contents of (**a**) 0.5%, (**b**) 1.0%, (**c**) 1.5%, (**d**) 2.0%, as PAA: PTFE–PAI@PEO was 5:5 and the sintering temperature was 400 °C; (**e**) Filtration efficiency for PM of different size; (**f**) the filtration efficiency and pressure drop for PM_2.0_.

**Table 1 nanomaterials-10-01806-t001:** The formula of PAI–PTFE–PEO solution used for electrospinning.

PAI–PTFE–PEO Nanofiber	Weight Ratio of Different Component				
	PAI	PTFE	PEO	H_2_O	PAI:PTFE
Hybrid–1	2.5	10	1	19	2:8
Hybrid–2	4.3	10	1	12	3:7
Hybrid–3	4.3	10	1	19	3:7
Hybrid–4	4.3	10	1	24	3:7
Hybrid–5	4.3	10	1	29	3:7
Hybrid–6	6.7	10	1	19	4:6
Hybrid–7	4.3	10	1	39	3:7
Hybrid–8	4.3	10	1	4	3:7

**Table 2 nanomaterials-10-01806-t002:** The filtration performance of PI/PTFE–PAI NAs of different fiber content as the ratio of PAA/PTFE was 5:5 and the sintering temperature was 400 °C.

Filtration Performance	Particle Size (μm)	Fiber Contents (%)			
		0.5%	1%	1.5%	2%
**Filtration Efficiency (%)**	≥0.3	81.21	87.20	95.36	97.35
	≥0.5	91.27	95.51	98.81	99.19
	≥1.0	97.63	99.45	99.87	99.88
	≥2.0	99.34	99.92	99.88	99.98
	≥5.0	99.60	99.89	99.59	99.98
	≥10.0	99.27	99.91	99.41	100
Pressure drop (Pa)	--	180	210	310	440

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
