# Peer review of "Preparation of PI/PTFE–PAI Composite Nanofiber Aerogels with Hierarchical Structure and High-Filtration Efficiency"

_nanomaterials, 2020, doi:10.3390/nano10091806_

Round 1

Reviewer 1 Report

In this paper, the preparation and characterization of nanofiber aerogels composites constructed with blocks of polytetrafluoroethylene-polyamideimide@polyethylene oxide composite nanofiber and polyamic acid (PAA) nanofiber are presented. The authors claim a hierarchical porous architecture and excellent mechanical properties for these materials due to thermally induced crosslink bonding. They have prepared composites with various porosity and shrinkage by adjusting the solid contents, nanofiber formula, and sintering temperature. As a main result, one of this composites could withstand high temperature up to 500 ℃ and an excellent performance in fine particulate matters filtration.

This is a paper full of details on the composites reported. This makes the lecture difficult and painful. All the composites all well described, both the preparation and the characterization. The nomenclature chosen to refer the different samples is hard. I have observed that some of the acronyms are mentioned without a previous identification.

The given errors interval for the fiber diameters are not the usual ones. One or two significant digits are adequate to represent the uncertainty. Two digits are given when these are below 25. The order of magnitude of this figure is that of the last figure of the measurement. Thus, 518±175 nm must be written (52±18)·10 and 341± 54 must be given (3,4±0,5)·10^2 (see, Taylor J. R. An Introduction to Error Analysis. The Study of Uncertainties in Physical Measurements. University Science Books (1982).

I do not find the importance of the term hierarchical, included in the title, regarding the filtration results. How many level of hierarchy are necessary to use it properly?

I do not find major objections for publications.

Author Response

#1 Comments and Suggestions for Authors

In this paper, the preparation and characterization of nanofiber aerogels composites constructed with blocks of polytetrafluoroethylene-polyamideimide@polyethylene oxide composite nanofiber and polyamic acid (PAA) nanofiber are presented. The authors claim a hierarchical porous architecture and excellent mechanical properties for these materials due to thermally induced crosslink bonding. They have prepared composites with various porosity and shrinkage by adjusting the solid contents, nanofiber formula, and sintering temperature. As a main result, one of this composites could withstand high temperature up to 500 ℃ and an excellent performance in fine particulate matters filtration.

  1. This is a paper full of details on the composites reported. This makes the lecture difficult and painful. All the composites all well described, both the preparation and the characterization. The nomenclature chosen to refer the different samples is hard. I have observed that some of the acronyms are mentioned without a previous identification.

Response: Thanks for your reviewing.

  • We are so sorry that there are still several defects in the manuscript. Following your advice, the discussion on the composites reported was shortened. The nomenclature for different sample was changed to a new set with a clear appearance. Also, the full names of the acronyms were added. More corrected details could be found in the revised manuscript.
  1. The given errors interval for the fiber diameters are not the usual ones. One or two significant digits are adequate to represent the uncertainty. Two digits are given when these are below 25. The order of magnitude of this figure is that of the last figure of the measurement. Thus, 518±175 nm must be written (52±18)·10 and 341± 54 must be given (3,4±0,5)·10^2 (see, Taylor J. R. An Introduction to Error Analysis. The Study of Uncertainties in Physical Measurements. University Science Books (1982).

Response: Thanks for your suggestion.

  • We are have transferred all the number of fiber diameter to the correct form with two significant digits. (Page 5, Line 177, Line 185)
  1. I do not find the importance of the term hierarchical, included in the title, regarding the filtration results. How many level of hierarchy are necessary to use it properly?

Response: Thanks for your suggestion.

  • More discussion about the importance of hierarchical structure was added in the manuscript (6 Section Page 14 Line 392), stating the filtration mechanism of the hierarchical structure. As for the levels of hierarchy, we think that more levels are always welcome. Generally, in the scale of micrometer, two levels of hierarchy are enough for the filtration of particles with a rather smaller pressure drop. If the roughness of the nanofiber surface could be improved (including porous surface, nano-rod on the surface, etc.), the capacity to capture fine particles would be promoted.
  1. I do not find major objections for publications.

Response: Thank you for your affirmation.

We will carefully check the manuscript and present you a faultless article.

Reviewer 2 Report

The manuscript “Preparation of PI/PTFE-PAI Composite Nanofiber Aerogels with Hierarchical Structure and High Filtration efficiency” deals with the production of polymeric filters, combining electrospinning with a deconstruction-reconstruction technique. Operating in this way, a filtration efficiency up to 99.91% for PM2.0 was obtained.

The work is interesting and numerous characterizations were performed on the samples. However, it can be improved highlighting some points.

-  Introduction does not point out the novelty of the work and the specific state of the art is missing. Please, add this information.

-  Results and discussion. Please, avoid to start directly the paragraph with a figure; add a brief description of the aim of those analyses, and, after that, show figures.

The reproducibility of this combination of techniques is not clear for the production of a porous aerogel network; please, discuss this aspect.

-  Conclusions. This paragraph has been written as a summary; some relevant information is missing, as for instance, the possibility of using this hybrid technique on large scale for a continuous production, generally used for this kind of products.

-  Please, check English and some typing errors in the text.

Author Response

Comments and Suggestions for Authors #2

The manuscript “Preparation of PI/PTFE-PAI Composite Nanofiber Aerogels with Hierarchical Structure and High Filtration efficiency” deals with the production of polymeric filters, combining electrospinning with a deconstruction-reconstruction technique. Operating in this way, a filtration efficiency up to 99.91% for PM2.0 was obtained.

The work is interesting and numerous characterizations were performed on the samples. However, it can be improved highlighting some points.

  1. Introduction does not point out the novelty of the work and the specific state of the art is missing. Please, add this information.

Response: Thanks for your suggestion.

  • The discussion on the novelty of this work was added in the last paragraph of Introduction section (Page 3, Line 93-104), giving a clear presentation of the novel strategy to prepare composite NAs.
  1. Results and discussion. Please, avoid to start directly the paragraph with a figure; add a brief description of the aim of those analyses, and, after that, show figures.

Response: Thanks for your suggestion.

  • A short introduction for each section was added in the revised manuscript, illustrating the necessity and function of each test. (Page 12 Line 344; Page13 Line 376; Page 14 Line392; Page15 Line424)
  1. The reproducibility of this combination of techniques is not clear for the production of a porous aerogel network; please, discuss this aspect.

Response: Thanks for your suggestion.

  • A detailed description of the preparation of the composite NAs network was presented in the revised manuscript. (Section 3.2 Fabrication of PI/PTFE-PAI NAs, Page7 Line 239~254)
  1. This paragraph has been written as a summary; some relevant information is missing, as for instance, the possibility of using this hybrid technique on large scale for a continuous production, generally used for this kind of products.

Response: Thanks for your suggestion.

  • The possible application was discussed in the Conclusion (Page 16 Line 467-469)
  1. Please, check English and some typing errors in the text.

Response: Thanks for your suggestion.

  • We are sorry for the mistakes in the text. A careful inspection has been conducted and those errors were amended, labeled in red in the revised manuscript.

Round 2

Reviewer 2 Report

Authors performed all changes proposed by the Reviewer. The manuscript can be accepted for publication.